# Non-pharmacologic hypertension management barriers and recommendations by hypertensive patients at Pentecost Hospital, Madina

**Evans Osei Appiah** [1]*, **Susana Boateng Agyekum**[2], **Amertil P. Ninon**[3], **Cyndi Appiah**[4]

**1** School of Nursing and Midwifery, Department of Midwifery, Valley View University, Accra, Ghana, **2** Nursing Department, School of Nursing and Midwifery, Valley View University, Accra, Ghana, **3** Dean of Nursing Department, School of Nursing and Midwifery, Valley View University, Accra, Ghana, **4** Assistant Lecturer, Ghana Christian University College, Accra, Ghana

* oseiappiahevans@ymail.com

**Data Availability Statement:** All data are in the manuscript and supporting information files.

**Funding:** The authors received no specific funding for this work.

## Abstract

The number of hypertension cases keeps rising worldwide. Africa is not exempted from the prevalence of hypertension. The Sub-Saharan region over the years has been recording high numbers of hypertension cases due to low consciousness, poor management and lack of control of urbanization. However, it has been established that hypertension as a condition can be managed by controlling familiar risk factors such as alcohol consumption, tobacco use, physical inactivity and intake of an unhealthy diet. The researchers, therefore, intend to explore the non-pharmacologic hypertension management barriers and recommendations by hypertensive patients at Pentecost Hospital, Madina. The researchers employed the qualitative exploratory-descriptive design using a purposive sampling technique to select 20 participants between the ages of 35–65, who met the inclusion criteria. Using a semi-structured interview guide, participants were engaged in 30–60 minutes of face-to-face interviews. The demography of the participants revealed that 60% (12) were females, and 40% (8) were also males. Participants reported that they visit the clinic once a week with a budget of hundred Ghana Cedis to five hundred Ghana Cedis (100–500 GHS). Two main themes and 7 subthemes emerged from the study analysis. The barriers identified include financial constraints, difficulty adjusting to lifestyle changes, personal factors (laziness, forgetfulness, stress), lack of motivation, and busy work schedules and limited time. Recommendations were also made to overcome the barriers which include follow ups by health care professionals, and advice to hypertensive and non-hypertensive patients. In conclusion, the study found that adherence to non-pharmacologic management of hypertension is greatly influenced by one's finances, some personal factors and external influences. Hence, it is necessary address these factors and also to ensure effective follow-ups and reminders in order to improve adherence to the non-pharmacologic management of hypertension. Further studies can also be conducted to address other obstacles to non-pharmacologic hypertension management.

**Competing interests:** The authors have declared that no competing interests exist.

## Introduction

Recently, the number of hypertension cases has alarmingly risen worldwide. For instance, the rate of hypertension incidence in England, USA, and Canada was 30%, 29%, and 19.5% respectively [1]. In this respect, it was noted that the average systolic blood pressure in England was higher than in the USA and Canada across all gender. Moreover, it was discovered in China that hypertension was on the rise as a total of 26.6% of adults were found to be with the condition [2]. However, this condition can be avoided by controlling familiar risk factors such as alcohol consumption, tobacco use, physical inactivity and intake of an unhealthy diet [3–5].

A study conducted in Africa revealed that there was a high prevalence of hypertension. The pervasiveness of the condition was attributed to factors such as low consciousness, poor management and control in the urban population [6]. In Egypt, it was estimated that a population of about 15 million people will be affected with hypertension by the year 2014. This was as a result of defective health care practitioners and deficient health care systems which served as obstacles in the management of hypertension [7]. He added that 60% of people with hypertension develop complications. He however confirmed that adequate provision of education by the physicians and workable healthcare facilities provided good channels in the management of hypertension. Nevertheless, it was established that only 25.8% of countries in Africa have developed guidelines for the effective management of hypertension.

In Ghana, it was discovered that hypertension incidence varied between 2.4% to 32.9% among people who were 15–19 years and 45–49 years [8]. It was however predicted that there will be a 60% increase in the year 2025 by 1.56 billion people in Ghana as compared to the year 2000 in which people with hypertension consisted of 972 million people [9]. Despite the increasing rate of hypertension in Ghana, poor control of hypertension was a familiar result [8, 10]. The poor control of hypertension in these studies was attributed to the deficiency of proper knowledge of hypertension, lack of skills and resources to manage and treat hypertension by some health systems, inadequate anti-hypertensive drugs, and long distances to the hospitals.

Patient's reluctance to adhere to lifestyle modification was identified as the major barrier to non-pharmacologic management of hypertension [11]. In Nepal, a study discovered that people consider lifestyle modification (diet) difficult despite their awareness of the numerous benefits associated with it [12]. The frequently reported barrier by the participants in this research was a food-related issue; the taste and desire for a certain food. It was also noted that dietary habits formed over a lifetime were difficult to break from [13]. Hence, the objective of this study was to assess non-pharmacologic hypertension management barriers to help recommend ways of addressing these barriers in order to improve adherence of hypertension patients to the non-pharmaclogic hypertensive management and to reduce hypertension burden locally and internationally.

## Methods

The researchers employed the qualitative exploratory-descriptive design to aid in exploring participants views on the perceived barriers, and make recommendations of non-pharmacological management of hypertension among hypertensive patients. This method and design also allowed participants to share their views on obstacles preventing them from adhering to the non-pharmacolgic hypertension management. A semi-structured interview guide was the data collection tool formulated by the researchers based on the study objectives and the literature review to provide insight into problems and broaden understanding regarding non-pharmacological hypertension management. See details of the interview guide in S1 File attached.

Hypertensive patients of Pentecost Hospital, Madina were the target population for the study. The hospital is located in the Greater Accra Region of Ghana, and its Out-Patient Department (OPD) runs four days a week, from 8 a.m to 2 p.m. However, the hypertensive clinic is organized on Thursdays and Fridays. The criteria for inclusion were; patients who had been diagnosed with hypertension for at least three (3) months. This group of hypertensive patients were considered based on shared common management methods of hypertension. In addition, the sampled population included hypertensive patients who were 35 years or more and were fluent speakers of English and Twi languages as these were languages spoken by the researchers. This was to ensure smooth communication between the researchers and the participants and also to present accurate information as the result of the study. Hypertensive patients who were seriously ill, unconscious and with other co-morbidities affecting their ability to speak were excluded from the study.

Participants were purposively selected with the study objective in mind to ensure that only participants who meets the inclusion criteria as stated above are selected and in other to ensure the richness of the data collected. The sample size was based on data saturation. Saturation is frequently used for determination of sample size qualitative studies [14]. Saturation is a sample determination size in qualitative research where responses are repeated by participants and data becomes redundant. The researchers kept recruiting and interviewing participants until the stage where no new data/information was retrieved (when participants started repeating the same ideas). This was reached on the 20th participant.

The data which were audio recordings of the interview sessions were transcribed and later analysed using a content analysis framework. In order to have total control over the data to facilitate coding and searching of the meaning and patterns, the transcript was read and reread. Codes that were accumulated were tabulated and grouped as themes and subthemes.

The methodological rigor of this study was maintain by ensuring that the data was credible and dependable. In addition, the confirmability and transferability were also maintained. This was ensured in the study by being cognizance of ethical issues, conducting a pilot study using 3 participants, describing the study design, sampling methods and technique. The interview guide was also provided and the whole study was done in accordance to the objectives of the study. This was done to ensure the trustworthiness of the study. Also, there was due diligence on the part of the researchers to guarantee the safety of the participants during the interview process. Privacy was maintained by interviewing respondents independently in a private office. Pseudonyms were accorded to each respondent to maintain confidentiality throughout the data collection. The respondents were also given a detailed explanation of the study, its methodology, its sample, data analysis and the procedural guidelines.

## Ethical consideration

The researchers obtained an ethical clearance and permission letter from the Dodowa Health Research Centre Institutional Review Board (DHRCIRB/35/03/20). The ethical clearance obtained was submitted to the leadership of Pentecost Hospital, Madina to prove the authenticity of the study. Following the approval from the authorities of the hospital, Participants were contacted after permission was sought from the ward in-charges. The participants were briefed on the purpose, as well as their role in the study. The sampled population was informed about the purpose of the study. Written consent forms were given to the participants to sign before the data collection procedure started. The participants were assured of confidentiality and were also given the privilege to ask any questions about the study. The recorded data were saved on the researchers' pen drive and laptop secured with a security code known to the researchers alone.

## Results

### Socio-demographic data

Twenty (20) participants with hypertension were interviewed. The participants were between the ages of thirty-five (35) and sixty-five (65). Thirteen (65%) were married, three (15%) were widows, and four (20%) were single. The English language was used for the interview. Ten (50%) participants were government workers and ten (50%) were self-employed (hairdresser, seamstress, shop-keeping, tailor). In terms of religion, the majority of the participants, 17 (85%) were from Christian denominations with three (15%) being Muslims. All the participants had some form of formal education:12 (60%) people have tertiary education, and 5 (25%) have only primary education. Participants resided in different locations in the Greater Accra Region of Ghana (Haatso, Adenta, Madina, Oyibi, Nungua, Amrahia, Legon, Frafraha, Shiashi, Amanfrom, and Okponglo). The demography of the participants revealed that females were 12, representing 60%, and the remaining40% (8) were males. The majority of the participants,75% reported that they visit the clinic once a week, with a budget ranged between one hundred and five hundred Ghana Cedis (100–500 GHS). The details of the socio-demographic characteristics of the participants are listed in Table 1.

### Themes and sub-themes

Two main themes and 7 sub-themes emerged from the study analysis. The two themes were perceived barriers of non-pharmacological management of hypertension, and recommendations by hypertensive patients. The themes and subthemes are shown in Table 2.

### Perceived barriers of non-pharmacological management of hypertension

There are five (5) sub-themes that emerged under this theme. These sub-themes—financial constraints, difficulty adjusting to lifestyle changes, personal factors (laziness, forgetfulness, stress), lack of motivation and busy work schedules and limited time–, our findings revealed, served as hindrances to the participants in their bid to follow a lifestyle modification regimen.

### Financial constraints

One of the barriers to following guidelines related to lifestyle recommendations was financial constraints usually accustomed to healthy foods. Below are excerpts from the interviews with the participants.

> *Economically, fruits and vegetables in the market are very expensive. It is not every day that you can buy fruits and vegetables to make a balanced meal. . . honestly, I think that if our income were a bit higher, we would be able to afford more balanced meals. Because it is expensive, sometimes I wish to follow a healthy pattern but I'm not able to–***(P20).**

> *It is actually not easy to eat healthy all the time, it is quite expensive. Sometimes, I try as much as possible to buy fruits and vegetables, but their prices scare me off. Even though I am single with no children to spend extra costs on, I'm unable to buy them. How much more having a family to cater for*? *I think I'll not be able to eat healthily all the time.–***(P13).**

It was also found that one's budget tends to increase especially when you have dependents.

> *mmm. . . the changes in diet in effect means spending more money. For example, staying in a household where you are supposed to cook for all of them. You can't cook for everyone the same way you cook for yourself because you are suffering from this disease. So, it means, you*

**Table 1. Socio-demographic characteristics of respondents.**

| Variable | Frequency (n = 20) | Per cent (%) |
|---|---|---|
| **Age group** | | |
| 18-35-40 | 5 | 25 |
| 21-41-50 | 2 | 10 |
| 22-51-59 | 3 | 15 |
| 26-60-65 | 10 | 50 |
| **Religion** | | |
| Christian | 17 | 85 |
| Muslim | 3 | 15 |
| Traditionalist | 0 | 0 |
| **Occupation** | | |
| Government worker | 10 | 50 |
| Self-employed | 10 | 50 |
| Not working | 0 | 0 |
| **Marital Status** | | |
| Single | 4 | 20 |
| Married | 13 | 65 |
| widow | 3 | 15 |
| **Educational status** | | |
| Primary | 5 | 25 |
| Secondary | 3 | 15 |
| Tertiary | 12 | 60 |
| **Gender** | | |
| Females | 12 | 60 |
| Males | 8 | 40 |
| Others | 0 | 0 |
| **Visit the clinic** | | |
| Once weekly | 15 | 75 |
| Twice weekly | 5 | 25 |
| Monthly | 0 | 0 |
| **Weekly budget** | | |
| 100–199 | 10 | 50 |
| 200–300 | 5 | 25 |
| 301–500 | 5 | 25 |
| 500 and above | 0 | 0 |

**Table 2. Themes and sub-themes.**

| THEMES | SUBTHEMES | CODES |
|---|---|---|
| **Barriers** | 1. financial constraints | **Bar** |
| | 2. difficulty adjusting to lifestyle changes | |
| | 3. personal factors (laziness, forgetfulness, stress) | |
| | 4. lack of motivation, and | |
| | 5. busy work schedules and limited time | |
| **Recommendations to overcome barriers** | 6. follow up by healthcare professionals | **Rec** |
| | 7. advice to hypertensive and non-hypertensive patients | |

*have to cook for them and cook yours later which will increase the budget that you would spend on the food.*–**(P6).**

### Difficulty adjusting to lifestyle changes

Lifestyle habits formed over a lifetime were difficult to change. Participants shared their challenging experiences in adopting new healthy lifestyle. They avowed that due to the difficult nature of adopting a new healthy lifestyle, they have always given up whenever they have made the attempt to engage in activities of such nature. Some of them shared their experiences as:

*Looking at the short amount of time you need to adjust to the changes, let's say, you've been eating a particular food for most of your life and in just one day, you are told to stop eating that particular food, it takes some time before you adjust, so if you're not careful, you might just say it's not anything and you might just go on with your previous life.*–**(P2).**

The study findings also revealed that hypertensive patients find difficulty in reducing or eliminating their meat intake because they have been taking meat throughout their lifetime. Below are the excerpts from such participants:

*Especially for my work, I am exposed to meat. I used to assist my mum who used to sell meat when we were young so it has almost become a part of me and I am finding it hard to reduce or stop eating it completely. I am really trying to do as the doctors have advised but is not easy.*–**(P9).**

*Well, I grew up in a home where the Ghanaian salted fish, I mean, "koobi", was used basically in most of the foods we prepared. It's actually impossible to remove it from the foods I prepare, especially beans stew. Anytime I am about to cook and I pick it up, I feel guilty, but I just can't put it down because it adds great taste to my food.*–**(P4).**

### Personal factors

From the interview conducted, participants mentioned that some personal factors that militate against successful non-pharmacological management of hypertension. Such personal factors include laziness to exercise, forgetfulness and stress due to busy work schedules and lack of discipline. They volitionally admitted that these issues are real factors obstructing a successful lifestyle modification. A few of the participants noted that they forget and usually feel lazy exercising all the time. Some responses from the participants are indicated below:

*Sometimes I forget to exercise. Sometimes, too, I just ignore due to body pains and laziness. Especially after a stressful day after work, it's hard to come home and exercise, much less take a stroll.*–**(P9).**

*In my case, I have to wake up early to exercise and prepare for work in the morning. I also have children which I have to take care of before work so sometimes if I feel lazy, I ignore the exercise and start with other activities.*–**(P6).**

Forgetfulness, stress and lack of discipline were also reckoned as barriers to lifestyle modification.

*I arrive home feeling tired and stressed from work most of the time and because of that, I am not able to prepare food so I resort to anything kept in the fridge.*–**(P11).**

*Sometimes I forget myself and I eat too much meat. Sometimes, the day will end before I realize I have not taken in any fruit.–***(P5).**

*It's difficult exercising because if you're not disciplined you can't follow a routine and may stop along the line. With the food too, looking at how appealing meat and drink are sometimes, it is usually difficult to control my appetite.–***(P5).**

### Lack of motivation

Motivation plays an important role in the treatment of hypertension. A few of the participants claimed that they feel relaxed when they are not motivated and hence, do not feel the need to engage in any lifestyle change.

*My doctor does not constantly call to check up on me to know about my progress as to whether I am being consistent or having any problems that may be delaying my recovery. It somehow makes me sad because it makes me think that he only cares when I go for the review. Due to that, I don't see the need to exercise or eat healthy regularly when I am at home because no one is monitoring me.–***(P15)**

Lack of support was also identified as one of the de-motivating factors to non-pharmacologic hypertension management.

*I am left alone to handle my condition myself since my family doesn't really care. It makes it hard especially when I have to cook two foods, one with salt and one without and they don't even support me. I can't do this all the time, so we all eat the same foods. I also don't exercise because I have no one to accompany me when I want to go on a stroll.–***(P19).**

### Busy work schedules and limited time

For some participants, the nature of their work serves as an obstacle to a successful lifestyle modification. Thus, they are unable to engage in activities of healthy living. In other words, their busy schedules do not permit them to embark on lifestyle changes models.

*Errm, some of the challenges I have so far is my work schedule. Because I am a trader, I order for some goods. However, some of the things that I order arrive late in the evenings so sometimes I work for half a night trying to offload and pack them. due to this, I arrive home very late and when that happens I am not able to eat early even though I might have already taken the medicine.–***(P11).**

*I get home late from work so I sometimes I end up eating late at night. And you know in Ghana, our heavy meals, like fufu is prepared in the evenings so it prevents me from sticking to the recommended guidelines.–***(P4).**

It was indicated that cooking consumes time and energy, as a result of that, participants resorted to buying food from outside. Consequently, some had little or no control over their diet and therefore made it difficult to engage in healthy lifestyle activities. The participant's assertion is indicated below:

*About my eating habits, I do not prepare my own food at home and since it requires more energy, I am forced to buy food from outside and sometimes the foods outside may not be healthy as compared the ones prepared home.–***(P2).**

*I wish to prepare all foods myself but I spend most of my time at work so I normally buy food from outside*–**(P18).**

## Recommendations

As a way of disabling the barriers to non-pharmacological management of hypertension, the participants recommended that there should be a follow-up service for hypertension patients by healthcare professionals, and also, there should be intensive public education on hypertension and non-pharmacological management of hypertension.

## Follow-up service by healthcare professionals

Even though some participants stated that they were often checked up by healthcare personnel, it was unfortunate that was not so for most of the participants. They, therefore, recommended that healthcare professionals should make it a point to frequently check up on them. They remarked as follows:

*Health workers should also check up on patients sometimes to see if they are doing well by following these changes. This is because some doctors just leave the patients after offering little education and they just leave them to fend for themselves.*–**(P8).**

*Errm, I think doctors and nurses should call or find ways to check up on us patients because it is very necessary. This is all for now.*–**(P11).**

*Eeerm, I think that patients diagnosed with Hypertension should be well educated on what it entails. There should be proper education which might in turn probably motivate us to follow the lifestyle changes so that we can manage our blood pressure from the doctors.*–**(P8).**

## Intensive public education on hypertension and non-pharmacological management of hypertension

Some participants also used this opportunity to advise their fellow hypertensive and non-hypertensive patients on preventive methods they could use in order to stay free from the condition. Below are some excerpts:

*I had the opportunity to prevent hypertension, but I chose not to listen. Now, I want to tell all those who do not have hypertension that, if possible, they should use salt moderately and try to reduce their intake of fatty foods. fruits and vegetables are something we Ghanaians are not fond of eating too. I think we all should make it a habit to start eating them.*–**(P6).**

Non hypertensive patients were also encouraged by the study participants to read a lot about ways to prevent hypertension.

*Before I was diagnosed, I had little interest in hypertension and the various ways to manage it. Due to this, I was unable to recognise the early signs of hypertension realized it was too late. So, I want to encourage those who have not been diagnosed or even those who have been diagnosed to frequently check their blood pressures to know the normal ranges and also practice the various lifestyle ways, such as reducing salt intake, exercising and sleeping well.*–**(P2).**

*Even if you have hypertension, you can have a semblance of a normal life even if you may not be able to do everything that everyone is doing. You are able to adjust your life to suit your current condition.*–**(P1).**

## Discussion

### Perceived barriers of non-pharmacological management of hypertension

In the present study, the results indicated that one of the key factors inhibiting changes in life-style behaviour towards management of hypertension was monetary consideration. The participants in this study assert that fruits and vegetables in the Ghanaian markets are costly nowadays. This in effect prevents them from adding fruits and vegetables to their diet daily. They indicated that even though they are encouraged to change their diet, their income levels serve as an impediment. This corresponds and affirms a study conducted in Eritrea which revealed that the pricey fruits are not patronized though they are necessities in the diet of hypertensive patients [15]. Admittedly, the study discovered that some fruits are low-priced, yet the participants claimed they are unable to afford them due to meagre salaries and financial constraints. This is of concern since non-pharmacological approach such as dietary modification was discovered to play a significant role in controlling blood pressure [16].

Moreover, it was realized from the study that making healthy living a lifestyle is a major problem for most people. Most of the participants in the study confirmed their uncomfortable experiences whenever they try to integrate the various lifestyle changes into their daily lives. Adapted to the habit of living a sedentary life, it was difficult trying to overturn their lives to live healthy overnight. Some also conferred that certain lifestyles–meat intake and intake of Ghanaian salted fish–could not be altered easily due to addiction. Any attempt they make to adopt a healthy lifestyle proves futile, hence their resolution to accept their current state. For instance, this finding resonates with a similar study in Barbados, which reported dietary habits formed over a period of time is very difficult to break from [16]. Thus, people have always encountered difficulties in trying to adopt lifestyle modifications that can help control, and manage hypertensive conditions. However, adhering to these lifestyle practices is of major benefits since some authors have established significant relationship between hypertension and the unhealthy lifestyle practices [3–5].

Personal factors such as non-adherence to an exercise plan, stress, and lack of discipline are accounted as barriers to lifestyle modification by some participants in the current study. Failure to exercise was the major personal factor that militates against successful non-pharmacological management of hypertension. This assertion is affirmed by the responses of the participants that they feel lazy to exercise due to body pains experienced from the previous day's exercises. There was also the claim that they sustained various injuries from the exercise they do, which discourages them from continuing [17]. Others also avow marathon of household chores render them exhausted, and make them forget to exercise. Furthermore, stress is another confirmed factor that prevents people from exercising. Stressed from the day's activities, participants mentioned that they are unable to exercise or eat healthily. According to a study conducted among African-Americans, stress is identified as a common barrier in managing hypertension [18]. This was mainly due to the busy nature of their daily activities. This was mainly due to the busy nature of their daily activities. Nevertheless, the importance of exercise in managing hypertension has been stressed on by several authors [19].

Another important factor in adhering to the lifestyle modification strategies of hypertension is motivation from health personnel or family. A couple of the participants unveiled that they were usually not motivated because they have no one to check up on them to ascertain their progress in adhering to the various health strategies. Importantly, they mentioned that apart from the weekly reviews they go for, they want the health personnel to check up on them and encourage them regularly to adhere to a healthy lifestyle regimen. The participants, therefore, maintained that lack of the needed encouragement prevented them from modifying their lifestyle to manage hypertension. Deficient health care systems and inadequate motivation by

the health practitioners discouraged participants in a study in Egypt as they stated that the channels the health facilities provided were not adequate in the management of hypertension [7]. Due to this, the health professionals failed to get in touch with their patients more often. Similarly, some researchers have revealed that hypertension in Ghana is poorly controlled [8, 10].

Time factor plays a significant role as a barrier to successful non-pharmacological hypertension management. Most of the participants of the study conferred that one major problem they have concerning hypertension management is time constraints. They posited that time constraints result in late and unhealthy eating. This also affects the time to exercise. To most of the participants, work has occupied all their time to the extent that they have limited time for their personal needs, including exercising. Research shows that thirty-nine per cent (39%) of clients in an interview pointed out that time constraints hinder their exercise schedules [19]. Other concerns also reveal that cooking consumes a lot of time and energy, and as a result that participants prefer eating from eateries to cooking their own food. Participants in the current study expressed this concern. Accumulation of these factors resulted in the failure of participants to modify their lifestyles in the management of hypertension. The results were also not surprising since a study in Nepal revealed that adherence to lifestyle modifications is a difficult behavior [12].

## Recommendations

Regular follow-up by health professionals serves as motivation for some participants. However, it is unfortunate that no conversations ensue between hypertensive patients and the health personnel in-between the review periods. Therefore, the participants suggest that the health personnel should frequently contact their patients in-between the review periods to serve as a follow-up service. Other participants also stated that the education they receive from the doctors were inadequate to help them manage hypertension, consequently leading to ineffective management of hypertension. It was also recommended that primary care providers should utilize a patient-centered approach in the care of patients with hypertension as their patients proved not to be familiar with the standard guidelines for managing hypertension, be it pharmacological or non-pharmacological [16]. A researcher also concluded that clinicians should spend adequate time in providing relevant information on the value of lifestyle modifications in controlling their blood pressure [20]. Thus, there should be intensive public education on non-pharmacological hypertension management.

The participants used in the current study seized the opportunity to reach out to both hypertensive patients and non-hypertensive patients, advising them to adhere to the various lifestyle strategies. They advised that people should, as a matter of importance, ensure preventive rather than curative methods in hypertension management. They reiterated that people should go for regular checkups in order to check their affordances. To the hypertensive patients, they encouraged them to have a normal life even though they may not resume their previous lifestyle. They added that hypertensive patients should try to adjust their lives to suit their current conditions in order to live to appreciate the life they have. It was however realized that the general public, unfortunately, has chosen not to heed any healthy lifestyle strategy until they are diagnosed. Some also have decided not to change their behaviors to suit their conditions, but rather, continued with their old unhealthy lifestyles [21, 22]. Similar to other findings, education, and provision of information has been identified as effective means in non-pharmacologic hypertension management adherence [6, 7].

## Conclusion

The study found that adherence to non-pharmacologic management of hypertension is greatly influenced by one's finances, some personal factors and external influences. Hence, it is

necessary address these factors and also to ensure effective follow-ups and reminders in order to improve adherence to the non-pharmacologic management of hypertension. Further studies can also be conducted to address other obstacles to non-pharmacologic hypertension management.

## Supporting information

**S1 File. Semi structured interview guide.**
(DOCX)

## Acknowledgments

The authors are very much appreciative to all authors whose work were cited in this study as well as the participants who availed themselves to partake in this study.

## Author Contributions

**Conceptualization:** Evans Osei Appiah, Susana Boateng Agyekum, Amertil P. Ninon, Cyndi Appiah.

**Data curation:** Evans Osei Appiah, Susana Boateng Agyekum, Amertil P. Ninon, Cyndi Appiah.

**Formal analysis:** Evans Osei Appiah, Susana Boateng Agyekum, Amertil P. Ninon.

**Investigation:** Evans Osei Appiah, Amertil P. Ninon, Cyndi Appiah.

**Methodology:** Evans Osei Appiah, Susana Boateng Agyekum, Amertil P. Ninon, Cyndi Appiah.

**Project administration:** Susana Boateng Agyekum, Amertil P. Ninon.

**Resources:** Evans Osei Appiah.

**Software:** Evans Osei Appiah.

**Validation:** Evans Osei Appiah, Amertil P. Ninon.

**Writing – original draft:** Evans Osei Appiah, Susana Boateng Agyekum, Amertil P. Ninon, Cyndi Appiah.

**Writing – review & editing:** Evans Osei Appiah, Susana Boateng Agyekum, Amertil P. Ninon, Cyndi Appiah.

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
