## [Decision Letter · Decision Letter 0]

5 Oct 2021

PGPH-D-21-00587

Non-pharmacologic hypertension management barriers and recommendations by hypertensive patients at Pentecost Hospital, Madina.

Dear Dr. Appiah,

Thank you for submitting your manuscript to PLOS Global Public Health. After careful consideration, we feel that it has merit but does not fully meet PLOS Global Public Health’s publication criteria as it currently stands. Therefore, we invite you to submit a revised version of the manuscript that addresses the points raised during the review process.

Please carefully consider the comments of reviewers #2 and the editorial comments and resubmit an updated version of your manuscript. 

We look forward to receiving your revised manuscript.

Kind regards,

Maurizio Trevisan, M.D., MS

Academic Editor

Journal Requirements:

1. Please provide separate figure files in .tif or .eps format only, and remove any figures embedded in your manuscript file.  If you are using LaTeX, you do not need to remove embedded figures.

2. Please provide a complete Data Availability Statement in the submission form, ensuring you include all necessary access information or a reason for why you are unable to make your data freely accessible. Note that it is not acceptable for the authors to be the sole named individuals responsible for ensuring data access.

PLOS defines a study's minimal data set as the underlying data used to reach the conclusions drawn in the manuscript and any additional data required to replicate the reported study findings in their entirety. Any potentially identifying patient information must be fully anonymized. 

If your research concerns only data provided within your submission, please write ""All data are in the manuscript and/or supporting information files"" as your Data Availability Statement.

Additional Editor Comments (if provided):

Please provide more detailed information on how the sample was selected and what was the participation rate. Especially in a study of small sample size (n=20) the process of selection and the participation rate play an important role in determining how representative the data are of the larger patients population

Please clarify what you mean by "were purposively selected using data saturation from the study setting"

Reviewers' comments:

Reviewer's Responses to Questions

**Comments to the Author**

1. Does this manuscript meet PLOS Global Public Health’s publication criteria? Is the manuscript technically sound, and do the data support the conclusions? The manuscript must describe methodologically and ethically rigorous research with conclusions that are appropriately drawn based on the data presented.

Reviewer #1: Yes

Reviewer #2: No

2. Has the statistical analysis been performed appropriately and rigorously?

Reviewer #1: Yes

Reviewer #2: N/A

3. Have the authors made all data underlying the findings in their manuscript fully available (please refer to the Data Availability Statement at the start of the manuscript PDF file)?

Reviewer #1: Yes

Reviewer #2: No

4. Is the manuscript presented in an intelligible fashion and written in standard English?

Reviewer #1: Yes

Reviewer #2: Yes

5. Review Comments to the Author

Reviewer #1: The manuscript written by the authors is very interesting and essential. We know that prevention and controlling of hypertension is a very big challenges always. However, the challenges is always remain under notices. The authors explore the barriers nicely by a qualitative study. However, my concern about the representativeness of the participant, otherwise all ae ok. Journal can consider for the publication.

Reviewer #2: Thank you for providing me an opportunity to review an article entitled “Non-pharmacologic hypertension management barriers and recommendations by hypertensive patients at Pentecost Hospital, Madina.” The article was designed qualitatively to explore the barriers for the non-pharmacological mgt of HTN. My comments on the paper are given below.

1. The abstract entered in the system and included in the paper is not similar. Please look into it for consistency.

2. Abstract – Results: A statement “Recommendations were also made to overcome the barriers” is not clear. Better if the authors list a summary of the actual recommendations mentioned.

3. Abstract – Conclusions: Participants mentioned several factors hindering their adherence to recommended nonpharmacological management of hypertension, and hence, recommended ways to help overcome these obstacles. This is broad and not inline with the study findings. Please clearly specify the conclusions based on findings.

4. Introduction: The rationale to conduct this study is not provided. Why you conducted this study?

5. The methods section is not well-written. The flow of ideas was not consistent. Some sections (e.g. ethical considerations) were repeated here and there. The design of the study is missing; what qualitative study design was used? Methods to ensure the trustworthiness of the qualitative study (credibility, transferability, dependability, confirmability) are not well described.

6. Results: Do not use words such as “some participants, one participant…”. Qualitative study is interested in ideas than the numbers.

7. Discussions: Lacks citations in some paragraphs and statements, no implications of the findings are provided.

8. Conclusions: are broad and not specific to the findings of the study.

9. Several grammatical errors are present in the article.

6. PLOS authors have the option to publish the peer review history of their article (what does this mean?). If published, this will include your full peer review and any attached files.

**Do you want your identity to be public for this peer review?** For information about this choice, including consent withdrawal, please see our Privacy Policy.

Reviewer #1: **Yes: **Palash Chandra Banik

Reviewer #2: No

---

## [Decision Letter · Decision Letter 1]

27 Oct 2021

PGPH-D-21-00587R1

Non-pharmacologic hypertension management barriers and recommendations by hypertensive patients at Pentecost Hospital, Madina.

Dear Dr. Appiah,

Thank you for submitting your manuscript to PLOS Global Public Health. After careful consideration, we feel that it has merit but does not fully meet PLOS Global Public Health’s publication criteria as it currently stands. Therefore, we invite you to submit a revised version of the manuscript that addresses the points raised during the review process.

We look forward to receiving your revised manuscript.

Kind regards,

Maurizio Trevisan, M.D., MS

Academic Editor

Journal Requirements:

Additional Editor Comments (if provided):

The authors have been responsive to the referees comments, however two issues remain to be clarified:

"The sample size was based on data saturation". Please clarify what is meant by this. I am not familiar with this terminology and the meaning of data saturation.

"The researchers kept recruiting and interviewing participants until the stage where no new data/information was retrieved. This was reached on the 20th participant."

It is important to specify the participation rate! How many patients were asked to participate in order to reach the require sample size? Knowing the participation rate is important to evaluate the value of the information retrieved.

Reviewers' comments:

Reviewer's Responses to Questions

**Comments to the Author**

1. If the authors have adequately addressed your comments raised in a previous round of review and you feel that this manuscript is now acceptable for publication, you may indicate that here to bypass the “Comments to the Author” section, enter your conflict of interest statement in the “Confidential to Editor” section, and submit your "Accept" recommendation.

Reviewer #1: All comments have been addressed

Reviewer #2: All comments have been addressed

2. Does this manuscript meet PLOS Global Public Health’s publication criteria? Is the manuscript technically sound, and do the data support the conclusions? The manuscript must describe methodologically and ethically rigorous research with conclusions that are appropriately drawn based on the data presented.

Reviewer #1: Yes

Reviewer #2: Yes

3. Has the statistical analysis been performed appropriately and rigorously?

Reviewer #1: Yes

Reviewer #2: N/A

4. Have the authors made all data underlying the findings in their manuscript fully available (please refer to the Data Availability Statement at the start of the manuscript PDF file)?

Reviewer #1: Yes

Reviewer #2: Yes

5. Is the manuscript presented in an intelligible fashion and written in standard English?

Reviewer #1: Yes

Reviewer #2: Yes

6. Review Comments to the Author

Reviewer #1: The manuscript is more improved now. Hypertension is a very big public health challenges at present and we found very little control patients. So, we need to know the reasons. I think the authors try appropriately to find out the reason.

Reviewer #2: The authors have satisfactorily addressed my previous comments.

7. PLOS authors have the option to publish the peer review history of their article (what does this mean?). If published, this will include your full peer review and any attached files.

**Do you want your identity to be public for this peer review?** For information about this choice, including consent withdrawal, please see our Privacy Policy.

Reviewer #1: **Yes: **Palash Chandra Banik

Reviewer #2: No

---

## [Editor Report · Decision Letter 2]

15 Nov 2021

Non-pharmacologic hypertension management barriers and recommendations by hypertensive patients at Pentecost Hospital, Madina.

PGPH-D-21-00587R2

Dear Dr. Appiah,

We're pleased to inform you that your manuscript has been judged scientifically suitable for publication and will be formally accepted for publication once it meets all outstanding technical requirements.

Within one week, you'll receive an e-mail detailing the required amendments. When these have been addressed, you'll receive a formal acceptance letter and your manuscript will be scheduled for publication.

An invoice for payment will follow shortly after the formal acceptance. To ensure an efficient process, please log into Editorial Manager at https://www.editorialmanager.com/pgph/ click the 'Update My Information' link at the top of the page, and double check that your user information is up-to-date. If you have any billing related questions, please contact our Author Billing department directly at authorbilling@plos.org.

Kind regards,

Maurizio Trevisan, M.D., MS

Academic Editor